# COVARIATE-INFORMED CONTINUOUS-TIME GRAY-BOX MODELING TO IDENTIFY RESPONSIVENESS OF POST-SURGICAL PAIN TO OPIOID THERAPY

## ABSTRACT

Quantifying responsiveness of pain to opioid administration is a clinically important, yet technically challenging problem. Pain is a subjective phenomenon that is difficult to assess by means other than infrequent and low-resolution patient self-reporting. We tackle this problem using a continuous-time state space modeling approach that incorporates mechanistic models of opioid effect site concentration as well as information from covariates using black-box models iteratively trained to predict the distributions of partially observed variables. We evaluated our method in simulation, and applied it in a real-world observational study of 21,652 surgical cases, where our method is able to recapitulate the known potencies of different opioids, and stratify patients by pain and opioid use related outcomes.

## 1 INTRODUCTION

Pain is an inherently subjective phenomenon, affected not only by physiological condition but also psychosocial factors (Lumley et al., 2011). For most practical purposes, pain can only be assessed through patient self-report, which yields infrequent, irregular, and low-resolution measurements (Williamson & Hoggart, 2005). Yet the assessment and management of pain is of critical importance to public health. Opioids are essential medicines for pain management, but growing rates of opioid dependence have led to a crisis (Rummans et al., 2018; Chisholm-Burns et al., 2019) that has expanded greatly over the past decades and is now responsible for over 80,000 deaths per year in the United States (Abuse, 2024). Tools to identify patients for whom opioids may be less effective for pain relief and that may have greater risk for dependence are greatly needed.

While reliance upon opioids for pain management has come under scrutiny (Hah et al., 2017), inadequately managed acute pain often becomes chronic and unresponsive to treatment (Glare et al., 2019), and non-opioid interventions do not provide the same degree of pain relief (Munir et al., 2007). Significant variability exists in responsiveness to opioid therapy due to difficult to assess factors such as individual variation in opioid receptor expression (Smith, 2008). We seek to quantify opioid responsiveness from observational data, to allow more objective and consistent clinical decision making.

Despite the myriad challenges posed both by the complexities of the underlying system and the available data, we propose a principled continuous-time state-space method for modeling the responsiveness of post-surgical pain to opioid therapy. We leverage known opioid pharmacokinetics and pharmacodynamics (Shafer et al., 1990; Shafer, 2014; Lamminsalo et al., 2019), as well as domain knowledge of the dependency structure between patient history and postoperative pain trajectories (Liu et al., 2023a), and provide a generalizable framework for training black-box models to predict latent state dynamics, enhancing our state space models with information from covariates.

We evaluated our method and its sensitivity to model misspecification in simulation, and applied it to real-world observational data from the first 24 hours after 21,652 surgical cases. Our method, with only an average of 16 pain observations across 24 hours per patient, is able to recapitulate known differences in potency between different opioids (Nielsen et al., 2016), and stratifies patients by outcomes associated with opioid responsiveness (Glare et al., 2019).

## 2 RELATED WORK

The task of quantifying response to intervention is often approached as a problem of causal inference (Pearl et al., 2016; Imbens & Rubin, 2015) and estimating individual treatment effects (Louizos et al., 2017). One of the most common approaches for estimating both average and individual treatment effects is by rebalancing covariates through back-door adjustment (Shalit et al., 2017).

However, achieving and verifying adequate covariate balance (Markoulidakis et al., 2021), as well as other conditions necessary for back-door adjustment, becomes increasingly challenging for time-varying exposures. For example, the overlap assumption (also referred to as positivity), which states that all covariate values have a nonzero probability of assignment to all exposure values, seems increasingly unlikely to hold as the dimensionality of covariates and exposures increases (D'Amour et al., 2021). In our specific problem of estimating opioid responsiveness, not all known confounders which influence pain and opioid administration, such as psychosocial factors, are observed. Nonetheless, others including Liu et al. (2023b) and Bica et al. (2020) have taken this approach by learning time-evolving covariate-balanced representations, and report good performance in terms of counterfactual prediction error within their modeling assumptions.

Zou et al. (2024) note that when modeling time-evolving systems, black-box models such as neural ODEs (Chen et al., 2018; Kidger, 2022) achieve better predictive performance than theory-based mechanistic models, but are more susceptible to confounding. A wide range of gray-box modeling (also referred to as hybrid model) approaches attempt to combine mechanistic and deep learning components (Wang & Fox, 2023; Hussain et al., 2021; Qian et al., 2021). Zou et al. (2024) provide a gray-box modeling framework which maintains the dependency relationships between states defined by a mechanistic model, and encodes knowledge about the expected direction of effect. Without constraints, a data-driven black-box model can simply cancel out a theory-informed model; Takeishi & Kalousis (2023) provide a regularization-based solution to this problem when fitting gray-box models.

Our approach is overall heavily constrained: we specify our state space models in parametric form, constrain our learned parameters by specifying the expected direction of effect, but incorporate information from covariates by using black-box model predictions of posterior distributions of parameters as priors.

## 3 METHODS

### 3.1 PROBLEM FORMULATION

For patient $j \in \{1, ..., N\}$, let $\mathbf{y_j} = [y_j(t_{j_1}), ..., y_j(t_{j_{n_j}})]$ such that $y_j(t_{j_i}) : \mathbb{R} \mapsto \{0, 1, ..., 10\}$ denotes their reported pain score at irregular observation times $t_{j_i}$; let $t_{j_0} < t_{j_1} < ... < t_{j_{n_j}}$ without loss of generality.

Let $\mathbf{u_j}(t) : \mathbb{R} \mapsto \mathbb{R}^m$ be their vector of opioid effect site concentrations over time, for $m$ different opioids. We estimate $\mathbf{u_j}(t)$ using existing pharmacokinetic/pharmacodynamic models parameterized by indvidual age, weight, height, and sex, as well as the timing and dosage of opioid boluses administered (Shafer et al., 1990; Lamminsalo et al., 2019). We used the open-source PK/PD simulation software Stanpump to estimate opioid effect site concentrations (ESC) based on these models (Shafer, 2014).

We define the following generative continuous-time state-space model for $\mathbf{y_j}$: let $x_j(t) : \mathbb{R} \mapsto \mathbb{R}$ be a patient's latent pain state at time $t$, with initial marginal distribution:

$$x_j(t_{j_0}) \sim \mathcal{N}(\mu_0, \sigma_0^2) \tag{1}$$

Let $\boldsymbol{x}_j = [x_j(t_{j_0}), x_j(t_{j_1}), ..., x_j(t_{j_{n_j}})]$ be their collection of latent states at initial time $t_{j_0}$ and times where $\boldsymbol{y}$ is observed.

Let $\mathbf{a_j} = [a_{j_1}, ..., a_{j_m}] \in \mathbb{R}_{\geq 0}^m$ be their vector of non-negative opioid response coefficients.

Let $\sigma^2$ be the variance of state noise over a time interval of length 1, and let $B_t$ be a Wiener process with variance 1 over a time interval of length 1.

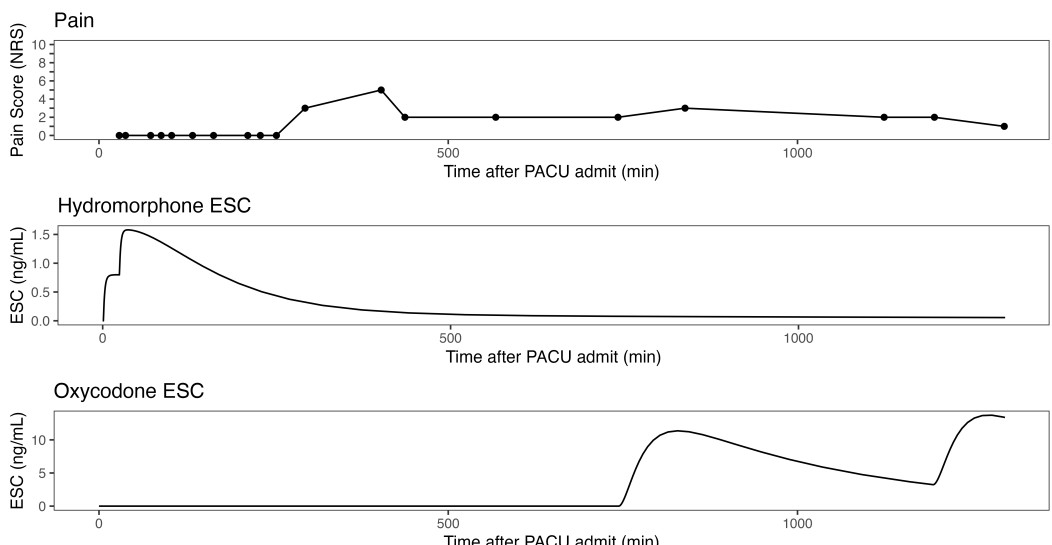

Figure 1: Pain scores: $\mathbf{y_j}$ and opioid effect site concentrations: $\mathbf{u_j}(t)$ over time for an example patient.

The evolution of $x_j(t)$ is then defined by the following stochastic differential equation (Kloeden et al., 1992):

$$dx_j(t) = -\mathbf{a_j} \cdot \mathbf{u_j}(t)dt + \sigma dB_t \tag{2}$$

Each $y_j(t_i)$ is then an ordered categorical random variable, also referred to as an ordinal random variable, with a total of $k = 11$ ordered categories $y_1 = 0, y_2 = 1, ..., y_k = 10$, and cumulative distribution function conditioned on $x(t_i)$ defined by:

$$P\left(y_j(t) \leq y_k \mid x_j(t)\right) = \frac{1}{1 + e^{x_j(t) - \beta_k}} \tag{3}$$

Observations of $\mathbf{y_j}$ and $\mathbf{u_j}(t)$ for one example patient are illustrated in Figure 1.

Finally, let $\mathbf{c_j} \in \mathbb{R}^l$ be a vector of covariates informative of initial state $x_j(t_{j_0})$ as well as opioid responsiveness $\mathbf{a_j}$. This is premised upon existing evidence that preoperative data is predictive of postoperative pain trajectory (Liu et al., 2023a) and is likely informative of opioid responsiveness. The dependencies between variables in our model are illustrated in Figure 2.

Our task is to rank patients by their opioid responsiveness parameter $\mathbf{a}_j$, given $\mathbf{y_j}, \mathbf{u_j}(t), \mathbf{c_j}$ for each patient. We evaluated the performance of our method in simulation, where ground truth for $\mathbf{a_j}$ is known. We also applied this method to observational data from 21,652 surgical patients at a quaternary care academic medical center in the United States, and examine associations between estimated $\mathbf{a}$ and postoperative outcomes associated with opioid responsiveness (Glare et al., 2019).

### 3.2 CONTINUOUS-TIME STATE SPACE MODELING WITH ORDINAL OBSERVATIONS

Typically, expectation-maximization procedures are used to fit state space models (McLachlan & Krishnan, 2007). When dynamics are linear and observations are Gaussian, Kalman filtering and fixed-interval smoothing can be used to estimate the posterior expectation $E[\mathbf{x_j} \mid \mathbf{a_j}, \mathbf{y_j}, \mathbf{u_j}(t)]$ (Shumway et al., 2000). However, because our observations $\mathbf{y_j}$ are non-Gaussian, there is no convenient analytic solution to the posterior.

Markov Chain Monte Carlo (MCMC) methods can be used instead to sample from the posterior distribution of $\mathbf{a_j}, \mathbf{x_j} \mid \mathbf{y_j}, \mathbf{u_j}(t)$. We can first derive a proposal function for a Metropolis-Hastings sampling scheme, which is proportional to the likelihood function of the distribution we are trying to sample, then use the No-U-Turn Sampler (NUTS) to efficiently sample from a Markov chain with an equivalent limiting distribution (Hoffman et al., 2014; Carpenter et al., 2017).

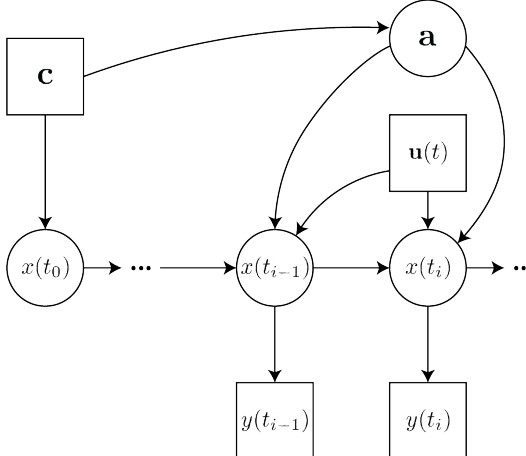

Figure 2: Dependency structure of our state space model. Variables which are observed are represented by squares, whereas variables which are unobserved are represented by circles.

Let $q(\mathbf{a}_j, \mathbf{x}_j)$ be our proposal function such that:

$$q(\mathbf{a}_j, \mathbf{x}_j) \propto \mathcal{L}\left(\mathbf{a}_j, \mathbf{x}_j \mid \mathbf{y}_j, \mathbf{u}_j(t)\right) \tag{4}$$

From Bayes' rule and the conditional independence properties of our model structure, we obtain (see Appendix A for a more thorough derivation):

$$\mathcal{L}\left(\mathbf{a}_j, \mathbf{x}_j \mid \mathbf{y}_j, \mathbf{u}_j(t)\right) = \frac{\mathcal{L}(\boldsymbol{y}_j \mid \boldsymbol{x}_j)\mathcal{L}\left(\boldsymbol{x}_j \mid \boldsymbol{a}_j, \boldsymbol{u}_j(t)\right)\mathcal{L}(a_j)\mathcal{L}\left(\boldsymbol{u}_j(t)\right)}{\mathcal{L}\left(\boldsymbol{y}_j, \boldsymbol{u}_j(t)\right)} \tag{5}$$

Factoring out all constants, with flat priors over $\mathbf{a}$, we have our proposal function:

$$q(\mathbf{a}_j, \mathbf{x}_j) = \mathcal{L}\left(x_j(t_{j_0})\right) \prod_{i=1}^{n_j} \mathcal{L}\left(y_j(t_{j_i}) \mid x_j(t_{j_i})\right) \prod_{i=1}^{n_j} \mathcal{L}\left(x_j(t_{j_i}) \mid x_j(t_{j_{i-1}}), \boldsymbol{a}_j, \boldsymbol{u}_j(t)\right) \tag{6}$$

We obtain $\mathcal{L}\left(y_j(t_{j_i}) \mid x_j(t_{j_i})\right)$ from Equation 3, and $\mathcal{L}\left(x_j(t_{j_0})\right)$ from Equation 1.

We can then solve Equation 2 over each time interval $[t_{j_{i-1}}, t_{j_i}]$. Our conditional state transition distributions over each interval give us $\mathcal{L}\left(x(t_{j_i}) \mid x(t_{j_{i-1}}), \boldsymbol{a}_j, \boldsymbol{u}_j(t)\right)$ and are given by:

$$x_j(t_{j_i}) \mid x_j(t_{j_{i-1}}), \mathbf{a}_j, \mathbf{u}_j(t) \sim \mathcal{N}\left(x_j(t_{j_{i-1}}) - \int_{t_{j_{i-1}}}^{t_{j_i}} \mathbf{a}_j \cdot \mathbf{u}_j(t)dt, \sigma\sqrt{t_{j_i} - t_{j_{i-1}}}\right) \tag{7}$$

Note that so long as we have analytic conditional state transition distributions, we only need to consider $x_j(t)$ at $t_{j_0}, t_{j_1}, ..., t_{j_{n_j}}$ which greatly reduces the number of variables which must be sampled compared to a discretized system.

Figure 3 illustrates the posterior distribution of $\boldsymbol{x}_j, \boldsymbol{a}_j \mid \boldsymbol{y}_j, \boldsymbol{u}_j(t)$ sampled using NUTS for the example patient in Figure 1. Point estimates for $\boldsymbol{a}_j$ are obtained for each patient by computing the mode of a kernel density estimate fitted to the samples (Sheather & Jones, 1991).

### 3.3 BLACK-BOX PREDICTION OF COVARIATE-INFORMED PRIORS

Now we wish to sample $\boldsymbol{a}_j, \boldsymbol{x}_j \mid \boldsymbol{y}_j, \boldsymbol{u}_j(t), \boldsymbol{c}_j$. From Bayes' rule and the conditional independence properties of our model, we obtain (see Appendix B for a more thorough derivation):

$$q(\boldsymbol{a}_j, \boldsymbol{x}_j, \boldsymbol{c}_j) = \mathcal{L}\left(x_j(t_{j_0}) \mid \boldsymbol{c}_j\right) \mathcal{L}\left(\boldsymbol{a}_j \mid \boldsymbol{c}_j\right) \prod_{i=1}^{n_j} \mathcal{L}\left(y(t_{j_i}) \mid x(t_{j_i})\right) \prod_{i=1}^{n_j} \mathcal{L}\left(x(t_{j_i}) \mid x(t_{j_{i-1}}), \boldsymbol{a}_j, \boldsymbol{u}_j(t)\right)$$

$$\tag{8}$$

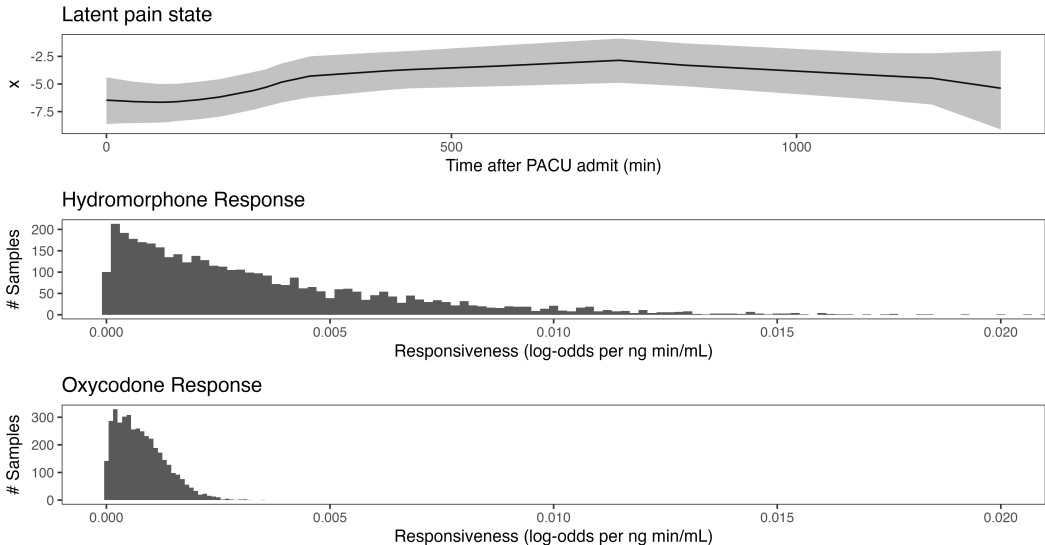

Figure 3: Posterior distributions of latent pain state: $\mathbf{x}_j \mid \mathbf{y}_j, \mathbf{u}_j(t)$ and opioid responsiveness: $\mathbf{a}_j \mid \mathbf{y}_j, \mathbf{u}_j(t)$ for the example patient shown in Figure 1. Shaded areas indicate 95% confidence intervals for $\mathbf{x}_j$. Responsiveness coefficients can be interpreted as log-odds of reduction in pain score per ng /mL of opioid effect site concentration per minute of exposure.

We now need models for $\mathcal{L}\left(x_j(t_{j_0}) \mid \mathbf{c}_j\right)$ and $\mathcal{L}\left(\mathbf{a}_j \mid \mathbf{c}_j\right)$, which we refer to as our covariate-informed priors. These can be black-box predictors; let $f, g$ be models with parameters $\theta_f$ and $\theta_g$.

$$\mathcal{L}\left(x_j(t_{j_0}) \mid \mathbf{c}_j\right) \approx \mathcal{L}\left(\hat{x}_j(t_{j_0}) \mid \mathbf{c}_j, \theta_f\right) = f(\mathbf{c}_j, \theta_f) \tag{9}$$

$$\mathcal{L}\left(\mathbf{a}_j \mid \mathbf{c}_j\right) \approx \mathcal{L}\left(\hat{\mathbf{a}}_j \mid \mathbf{c}_j, \theta_g\right) = g(\mathbf{c}_j, \theta_g) \tag{10}$$

Our training objective is to minimize the total Kullback-Leibler divergence between our covariate-informed priors and our posterior distributions, across all patients.

$$\theta_f^* = \arg\min_{\theta_f} \sum_j^N D_{KL}\left((x_j(t_{j_0}) \mid \mathbf{y}_j, \mathbf{u}_j(t), \mathbf{c}_j), (\hat{x}_j(t_{j_0}) \mid \mathbf{c}_j, \theta_f)\right) \tag{11}$$

$$\theta_g^* = \arg\min_{\theta_g} \sum_j^N D_{KL}\left((\mathbf{a}_j \mid \mathbf{y}_j, \mathbf{u}_j(t), \mathbf{c}_j), (\hat{\mathbf{a}}_j \mid \mathbf{c}_j, \theta_g)\right) \tag{12}$$

For computational convenience, we approximate the posterior distributions of $x_j(t_{j_0})$ as Gaussian, and $\mathbf{a}_j$ as log-Gaussian with variance constant with respect to $\mathbf{c}_j$, which reduces the minimization of KL-divergence to least-squares on the posterior means of each patient (see Appendix C).

However, $x_j(t_{j_0})$ and $\mathbf{a}_j$ are not directly observed, and so we cannot simply perform supervised learning. Instead, we devise an expectation-maximization algorithm (Dempster et al., 1977; McLachlan & Krishnan, 2007):

In the first E-step, we begin with flat priors on $\mathbf{a}_j$, and $x_j(t_{j_0})$ independent of $\mathbf{c}_j$, using Equation 6 to sample from the posterior distributions of $x_j(t_{j_0})$ and $\mathbf{a}_j$. In each M-step, we then update $\theta_f$ and $\theta_g$ using Equation 11 and 12 with the previous E-step's sample estimates. In subsequent E-steps, using the previous M-step's learned values of $\theta_f, \theta_g$, we then update the posterior distributions of $x_j(t_{j_0}), \mathbf{a}_j \mid \mathbf{y}_j, \mathbf{u}_j(t), \mathbf{c}_j, \theta_f, \theta_g$ using Equation 8. On iteration $T$, this is equivalent to:

$$\theta_f^{(T+1)} = \arg\min_{\theta_f} \sum_j^N D_{KL}\left(\left(\hat{x}_j(t_{j_0}) \mid \mathbf{y}_j, \mathbf{u}_j(t), \mathbf{c}_j, \theta_f^{(T)}\right), (\hat{x}_j(t_{j_0}) \mid \mathbf{c}_j, \theta_f)\right) \tag{13}$$

$$\theta_g^{(T+1)} = \arg\min_{\theta_g} \sum_j^N D_{KL}\left(\left(\hat{\boldsymbol{a}}_j \mid \boldsymbol{y}_j, \boldsymbol{u}_j(t), \boldsymbol{c}_j, \theta_g^{(T)}\right), \left(\hat{\boldsymbol{a}}_j \mid \boldsymbol{c}_j, \theta_g\right)\right) \tag{14}$$

We repeat this procedure until a pre-specified number of iterations has been reached, or the total data log-likelihood is no longer increasing.

### 3.4 SIMULATION STUDY

We generated simulated data as numerical realizations of Equation 2 for varying values of $\sigma$. Covariates $\boldsymbol{c}$ were generated as realizations of independent Gaussian random variables such that a prespecified proportion $r^2$ of the variance in $x_{j_0}$ and $\boldsymbol{a}$ were explained by their sum.

Wherever possible, parameters were chosen to match distributions found in our observational dataset, including the frequency of observations and opioid administration, as well as patient demographics (further details in Appendix D). $\boldsymbol{y}_j$ was observed according to Equation 3 with exponentially distributed intervals between observations.

We applied the same data exclusion criteria to our simulated data as in our observational study; we rejected data for which model identification was impossible, such as patients with only 0 or 10 pain scores, or patients who did not receive opioids.

We considered cases in which varying fractions of variance $r^2$ in $x_j(t_{j_0})$ and $\boldsymbol{a}_j$ are explained by $\boldsymbol{c}_j$, comparing performance metrics with and without covariate-informed priors as an ablation study. We compute concordance probability (also referred to as c-index, and equal to AUC when one variable is binary) between true $\boldsymbol{a}_j$ and sampled posterior mode, as well as Kendall's rank correlation ($\tau$) in order to assess the quality of our estimated rankings. Confidence intervals for our performance metrics were computed using bootstrap.

### 3.5 REAL-WORLD APPLICATION ON OBSERVATIONAL DATA

We applied our method to electronic health record data from 21,652 adult surgical patients at a quaternary care academic medical center in the United States, computing responsiveness of postoperative pain to fentanyl, hydromorphone, and oxycodone. Our protocol was approved by the Institutional Review Board with a waiver of informed consent. Covariates in this dataset consisted of routinely collected preoperative data, including patient demographics, surgical service of their scheduled procedure, surgery urgency (elective, urgent, emergent), preoperative pain score, opioid naivety, and other elements of patient medical and surgical history. We included only patients who had at least 10 pain score observations in the first 24 hours after surgery, and were administered hydromorphone or oxycodone at least once after surgery. Carry-over from intraoperative opioid administration was factored into our PK/PD calculations; fentanyl was only administered intraoperatively or immediately after surgery in the PACU.

Patients were stratified into high and low responsiveness cohorts for hydromorphone and oxycodone based on whether their estimated responsiveness was greater or less than the median in our study cohort; the distribution of postoperative outcomes was evaluated within these cohorts.

## 4 RESULTS

### 4.1 SIMULATION STUDY

Table 1 gives performance of our continuous-time state space modeling method on simulated data for different values of $\sigma$, the magnitude of noise in the state evolution equation, without informative covariates $\boldsymbol{c}$. Higher is better for both concordance probability (C-index) and Kendall's rank correlation ($\tau$). The general trend observed is that the system becomes less identifiable as the magnitude of noise in the state evolution equation increases. Without informative covariates, sample size does not impact the performance of our model, only the width of the confidence bounds of our performance metrics, as each patient's parameters are estimated independently of the others.

Table 2 gives performance of our method on simulated data for varying values of $r^2$, the percentage of variance in $x_{j_0}$ and $\boldsymbol{a}_j$ explained by $\boldsymbol{c}_j$, and compares performance with flat priors (without $\boldsymbol{c}$),

Table 1: Parameters and performance metrics with 95% confidence intervals for simulation study

| $N$ | $\sigma$ | C-index | Kendall's $\tau$ |
|---|---|---|---|
| 1000 | 0.1 | .785 (.771, .798) | .569 (.543, .595) |
| 1000 | 0.2 | .718 (.701, .735) | .437 (.405, .470) |
| 1000 | 0.3 | .670 (.651, .688) | .340 (.302, .377) |
| 1000 | 0.4 | .653 (.633, .673) | .306 (.267, .344) |
| 1000 | 0.5 | .634 (.614, .653) | .267 (.230, .303) |
| 1000 | 0.6 | .607 (.587, .626) | .214 (.175, .253) |
| 1000 | 0.7 | .599 (.579, .619) | .198 (.158, .238) |
| 1000 | 0.8 | .597 (.577, .616) | .195 (.154, .237) |

and with covariate-informed priors (with $c$). We find that while covariates that provide information about the distribution of the initial state $x_{j_0}$ may improve the performance of our method when $c$ is informative of $a$, when $c$ is not informative of $a$, there is no performance gain when $c$ is not informative of $a$. Only covariates that can predict $a$ improve our ability to identify opioid responsiveness; the more informative $c$ is of $a$, the greater the increase in performance. We also find that the performance increase from using covariate-informed priors is slightly greater for lower $\sigma$. This is likely because if the overall system is more identifiable, then the black box predictors will be trained on more accurate labels, thus increasing the amount of information that they provide.

Table 2: Parameters and performance metrics with 95% confidence intervals for simulation and ablation study

| | | | | with $c$ | | without $c$ | |
|---|---|---|---|---|---|---|---|
| $N$ | $r_{x_0}^2$ | $r_a^2$ | $\sigma$ | C-index | Kendall's $\tau$ | C-index | Kendall's $\tau$ |
| 2000 | 1 | 1 | 0.1 | .851 (.845, .858) | .703 (.690, .716) | .777 (.768, .786) | .555 (.535, .573) |
| 2000 | 1 | 1 | 0.2 | .788 (.780, .797) | .577 (.558, .595) | .716 (.704, .727) | .432 (.409, .454) |
| 2000 | 1 | 1 | 0.3 | .739 (.728, .750) | .478 (.454, .499) | .675 (.662, .687) | .349 (.325, .374) |
| 2000 | 0 | 1 | 0.3 | .729 (.716, .741) | .458 (.433, .480) | .673 (.660, .685) | .345 (.318, .370) |
| 2000 | 1 | 0 | 0.3 | .683 (.670, .695) | .366 (.341, .391) | .680 (.668, .693) | .361 (.334, .386) |
| 2000 | 0.5 | 0.5 | 0.3 | .699 (.686, .712) | .398 (.373, .424) | .674 (.661, .687) | .348 (.323, .372) |
| 2000 | 0.5 | 0 | 0.3 | .692 (.679, .705) | .384 (.358, .410) | .669 (.655, .682) | .338 (.311, .363) |
| 2000 | 0 | 0.5 | 0.3 | .671 (.657, .683) | .341 (.314, .367) | .671 (.658, .684) | .341 (.313, .369) |
| 2000 | 0.3 | 0.3 | 0.3 | .689 (.677, .702) | .378 (.351, .404) | .673 (.659, .685) | .346 (.319, .373) |
| 2000 | 0.2 | 0.2 | 0.3 | .677 (.665, .690) | .355 (.327, .380) | .663 (.650, .676) | .325 (.298, .351) |

## 4.2 REAL-WORLD APPLICATION ON OBSERVATIONAL DATA

Our observational cohort consisted of 21,652 adult non-cardiac surgical cases with general anesthesia. On average, there were 16.3 pain score observations, 2.2 instances of postoperative hydromorphone administration, and 2.6 instances of postoperative oxycodone administration per patient. Further demographics and study cohort characteristics are reported in Table 4.

Figure 4 shows the distribution of estimated opioid responsiveness parameters for fentanyl, hydromorphone, and oxycodone across our study cohort. The ratios between these values appear comparable to literature equianalgesic dose ratios between fentanyl, hydromorphone, and oxycodone (Pereira et al., 2001). The ratio between our median estimated fentanyl and hydromorphone responsiveness parameters is 4.9, whereas the literature ratio of equivalent dosages is 7.5; moreover, fentanyl is known to have a shorter duration of action compared to hydromorphone, and our estimated responsiveness is per duration of exposure. The ratio between our median estimated hydromorphone and oxycodone responsiveness is 12.8, whereas the literature reports ratios between 2.0 and 2.7.

Table 3 gives distributions of postoperative outcomes for patients whose estimated hydromorphone and oxycodone responsiveness fell into the top half (high responsiveness) or bottom half of the

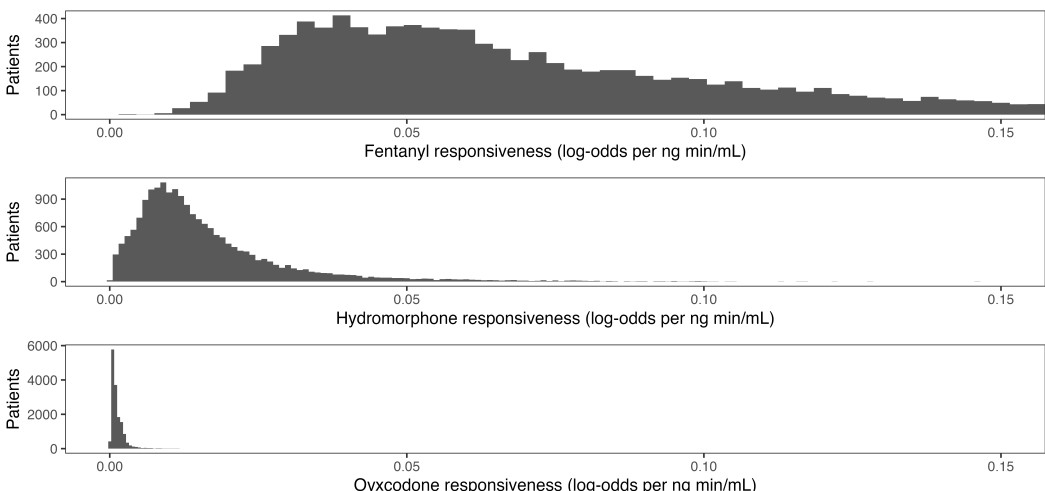

Figure 4: Study cohort distribution of responsiveness for fentanyl, hydromorphone, and oxycodone.

study cohort. We find that high responsiveness of postoperative pain to both hydromorphone and oxycodone is associated with better overall outcomes, including lower postoperative pain, lower total opioid usage in hospital, lower rates of postoperative opioid prescriptions, lower rates of chronic pain diagnosis, lower rates of readmission, and shorter hospital stays.

Table 3: Postoperative outcomes for patients stratified by opioid responsiveness.

| Outcome | Hydromorphone | | Oxycodone | |
|---|---|---|---|---|
| | Low | High | Low | High |
| N | 7,685 | 7,684 | 8,119 | 8,118 |
| Max Pain in PACU | 7 (5, 9) | 6 (5, 8) | 7 (5, 9) | 5 (3, 7) |
| Max Pain 24h Postop | 8 (7, 10) | 7 (6, 9) | 8 (7, 10) | 7 (6, 8) |
| Max Pain in Hospital | 9 (8, 10) | 8 (6, 9) | 9 (8, 10) | 7 (6, 9) |
| Total MME in PACU | 6.7 (3.4, 11.7) | 4.2 (3.4, 6.7) | 6.7 (3.4, 11.7) | 3.4 (0.0, 6.7) |
| Total MME 24h Postop | 24.2 (15.1, 36.7) | 12.0 (6.7, 20.9) | 35.3 (20.0, 36.7) | 10.0 (5.9, 14.9) |
| Total MME in Hospital excl. PACU | 33.4 (14.2, 83.8) | 11.7 (2.5, 30.0) | 41.7 (21.3, 89.8) | 10.0 (5.0, 22.5) |
| 30-day Opioid Rx | 43.9% | 32.9% | 45.2% | 32.9% |
| 90-day Opioid Rx | 46.3% | 34.5% | 47.5% | 34.5% |
| 180-day Opioid Rx | 47.4% | 35.6% | 48.6% | 35.8% |
| 3-month Chronic Pain Dx | 0.83% | 0.68% | 0.71% | 0.62% |
| 12-month Chronic Pain Dx | 3.19% | 2.58% | 3.11% | 2.22% |
| 30-day Readmission | 12.02% | 8.49% | 11.37% | 9.13% |
| Hospital Length of Stay (hrs) | 74.4 (34.9, 143.0) | 52.5 (30.2, 99.6) | 74.2 (36.6, 138.1) | 52.1 (29.6, 99.9) |

## 5 DISCUSSION

We present a method for estimating responsiveness of postsurgical pain to opioid therapy using observations of patient-reported pain scores over time, medication records, and preoperative covariates. We evaluated some of the performance characteristics of this method in simulation, and applied it to observational data from real patients. Our method was able to recapitulate the known relative potency of different opioids, and was able to stratify patients by postoperative outcomes related to pain and opioid usage based on their estimated responsiveness to hydromorphone and oxycodone. Moreover, our model-estimated responsiveness is interpretable in terms of physical quantities: for

each opioid, we have the log-odds of reduction in observed pain score per minute of exposure to a given effect site concentration.

Another strength of this method is that it is uncertainty-aware; rather than simply producing a point estimate of responsiveness for each patient, we estimate a posterior distribution. The uncertainty in the distribution of a patient's latent pain state increases with the duration of the interval between observations. Patients with sparse data will have posteriors with greater variance. Our method is also able to take into account information from covariates that purely mechanistic models cannot, without explicitly specifying the relationship between those variables and the dynamical system, while preserving causal relationships and mechanistic structures between variables.

While our model was designed with the specific application of estimating postoperative opioid responsiveness in mind, and consequently is able to leverage known opioid PK/PD and dependency structures, our general concept of using black-box models to augment state space models with covariate-informed priors is potentially applicable to other gray-box modeling problems, data modalities, and network structures.

## 5.1 LIMITATIONS

While using MCMC to estimate posterior distributions affords us flexibility in the types of observations we are able to use in our state space models and makes it easy to incorporate priors, a consequence of this approach is a relatively long run time. In our dataset, on a 16-core AMD 5955WX CPU, it takes about 2 seconds per patient to run one E-step, which equates to about 72 hours to analyze our dataset of over 20,000 patients. Overall, these computations remain tractable because we are sampling a relatively small number of parameters; however, that also means that GPU-based MCMC implementations do not provide any performance increase. Moreover, because our E-step is not analytic and contains sampling noise, our EM procedure is not guaranteed to converge.

Furthermore, we assume that opioid responsiveness is time-invariant. However, this is not necessarily true even within the 24 hour time window which we consider, as tolerance can develop on the order of hours (Dumas & Pollack, 2008), and there are also other mechanisms of sensitization that could modulate opioid responsiveness over time (Woolf, 2011). This could contribute to the discrepancy we observe between our estimated hydromorphone and oxycodone responsiveness and literature-reported ratios of equivalent doses, as oxycodone tends to be administered later in perioperative care than hydromorphone and fentanyl.

Finally, we have a single set of intercepts $\beta$ for our observations generated by Equation 3. In order for us to compare responsiveness of pain to opioid administration between patients, it is necessary that their pain states evolve in a common latent space. We treat the 0-10 numerical pain scale as fixed across all patients, even though in reality, different patients may report different scores after experiencing the same degree of painful stimulus. However, violation of this assumption is not necessarily fatal, so long as we shift our interpretation of estimated opioid responsiveness to a kind of average odds ratio across the different levels of the pain scale (Harrell, 2020).

## REPRODUCIBILITY STATEMENT

An anonymous archive containing source code for all methods and for running experiments with simulated data has been included with this submission. Access to clinical data used in this paper is subject to a data use agreement with the institution where data was collected. Once this data use agreement is obtained, the authors are able to share the clinical data for the real-world portion of this study.

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

## A  METROPOLIS-HASTING PROPOSAL FUNCTION

$$\mathcal{L}\left(\boldsymbol{a}_j, \boldsymbol{x}_j \mid \boldsymbol{y}_j, \boldsymbol{u}_j(t)\right) = \frac{\mathcal{L}\left(\boldsymbol{y}_j \mid \boldsymbol{a}_j, \boldsymbol{x}_j, \boldsymbol{u}_j(t)\right)\mathcal{L}\left(\boldsymbol{a}_j, \boldsymbol{x}_j, \boldsymbol{u}_j(t)\right)}{\mathcal{L}\left(\boldsymbol{y}_j, \boldsymbol{u}_j(t)\right)} \tag{15}$$

From Figure 2, we have that $\boldsymbol{y}_j$ is conditionally independent of $\boldsymbol{a}_j$ and $\boldsymbol{u}_j(t)$, given $\boldsymbol{x}_j$.

$$\mathcal{L}\left(\boldsymbol{y}_j \mid \boldsymbol{a}_j, \boldsymbol{x}_j, \boldsymbol{u}_j(t)\right) = \mathcal{L}\left(\boldsymbol{y}_j \mid \boldsymbol{x}_j\right) = \prod_{i=1}^{n_j}\mathcal{L}\left(y_j(t_{j_i}) \mid x_j(t_{j_i})\right) \tag{16}$$

$$\mathcal{L}\left(\boldsymbol{a}_j, \boldsymbol{x}_j, \boldsymbol{u}_j(t)\right) = \mathcal{L}\left(\boldsymbol{x}_j \mid \boldsymbol{a}_j, \boldsymbol{u}_j(t)\right)\mathcal{L}\left(\boldsymbol{a}_j, \boldsymbol{u}_j(t)\right) \tag{17}$$

We also have that $x_{j_0}$, $\boldsymbol{a}_j$, and $\boldsymbol{u}_j(t)$ are independent.

$$\mathcal{L}\left(\boldsymbol{a}_j, \boldsymbol{u}_j(t)\right) = \mathcal{L}(\boldsymbol{a}_j)\mathcal{L}\left(\boldsymbol{u}_j(t)\right) \tag{18}$$

$$\mathcal{L}\left(\boldsymbol{x}_j \mid \boldsymbol{a}_j, \boldsymbol{u}_j(t)\right) = \mathcal{L}(x_{j_0})\prod_{i=1}^{n_j}\mathcal{L}\left(x_j(t_{j_i}) \mid x_j(t_{j_{i-1}}), \boldsymbol{a}_j, \boldsymbol{u}_j(t)\right) \tag{19}$$

Finally, we factor out $\mathcal{L}(\boldsymbol{a}_j)$, $\mathcal{L}(\boldsymbol{u}_j(t))$, and $\mathcal{L}\left(\boldsymbol{y}_j, \boldsymbol{u}_j(t)\right)$ as constants. This gives us Equation 6.

## B  METROPOLIS-HASTING PROPOSAL FUNCTION WITH COVARIATE-INFORMED PRIORS

$$\mathcal{L}\left(\boldsymbol{a}_j, \boldsymbol{x}_j \mid \boldsymbol{y}_j, \boldsymbol{u}_j(t), \boldsymbol{c}_j\right) = \frac{\mathcal{L}\left(\boldsymbol{y}_j \mid \boldsymbol{a}_j, \boldsymbol{x}_j, \boldsymbol{u}_j(t), \boldsymbol{c}_j\right)\mathcal{L}\left(\boldsymbol{a}_j, \boldsymbol{x}_j, \boldsymbol{u}_j(t), \boldsymbol{c}_j\right)}{\mathcal{L}\left(\boldsymbol{y}_j, \boldsymbol{u}_j(t), \boldsymbol{c}_j\right)} \tag{20}$$

$\boldsymbol{y}_j$ is conditionally independent of $\boldsymbol{a}_j$, $\boldsymbol{u}_j(t)$, and $\boldsymbol{c}_j$, given $\boldsymbol{x}_j$.

$$\mathcal{L}\left(\boldsymbol{y}_j \mid \boldsymbol{a}_j, \boldsymbol{x}_j, \boldsymbol{u}_j(t), \boldsymbol{c}_j\right) = \mathcal{L}\left(\boldsymbol{y}_j \mid \boldsymbol{x}_j\right) = \prod_{i=1}^{n_j}\mathcal{L}\left(y_j(t_{j_i}) \mid x_j(t_{j_i})\right) \tag{21}$$

$$\mathcal{L}\left(\boldsymbol{a}_j, \boldsymbol{x}_j, \boldsymbol{u}_j(t), \boldsymbol{c}_j\right) = \mathcal{L}\left(\boldsymbol{x}_j \mid \boldsymbol{a}_j, \boldsymbol{u}_j(t), \boldsymbol{c}_j\right)\mathcal{L}\left(\boldsymbol{a}_j, \boldsymbol{u}_j(t), \boldsymbol{c}_j\right) \tag{22}$$

We have that $\boldsymbol{u}_j(t)$ is independent of $\boldsymbol{c}_j$ and $\boldsymbol{a}_j$, and $x_{j_0}$ and $\boldsymbol{a}_j$ are conditionally independent given $\boldsymbol{c}_j$.

$$\mathcal{L}\left(\boldsymbol{a}_j, \boldsymbol{u}_j(t), \boldsymbol{c}_j\right) = \mathcal{L}\left(\boldsymbol{a}_j \mid \boldsymbol{c}_j\right)\mathcal{L}\left(\boldsymbol{u}_j(t)\right)\mathcal{L}(\boldsymbol{c}_j) \tag{23}$$

$$\mathcal{L}\left(\boldsymbol{x}_j \mid \boldsymbol{a}_j, \boldsymbol{u}_j(t), \boldsymbol{c}_j\right) = \mathcal{L}(x_{j_0} \mid \boldsymbol{c}_j)\prod_{i=1}^{n_j}\mathcal{L}\left(x_j(t_{j_i}) \mid x_j(t_{j_{i-1}}), \boldsymbol{a}_j, \boldsymbol{u}_j(t)\right) \tag{24}$$

Finally, we factor out $\mathcal{L}(\boldsymbol{c}_j)$, $\mathcal{L}(\boldsymbol{u}_j(t))$, and $\mathcal{L}\left(\boldsymbol{y}_j, \boldsymbol{u}_j(t), \boldsymbol{c}_j\right)$ as constants. This gives us Equation 8.

## C  MINIMIZATION OF KL DIVERGENCE

The KL divergence between two Gaussian distributions $P \sim \mathcal{N}(\mu_p, \sigma_p^2)$, $Q \sim \mathcal{N}(\mu_q, \sigma_q^2)$ is given by (Soch, 2020):

$$D_{KL}(P \mid Q) = \frac{1}{2}\left(\frac{(\mu_p - \mu_q)^2}{\sigma_q^2} + \frac{\sigma_p^2}{\sigma_q^2} - \ln\frac{\sigma_p^2}{\sigma_q^2} - 1\right) \tag{25}$$

For patient $j$, let $P_j \sim \mathcal{N}(\mu_{p_j}, \sigma_{p_j}^2)$ denote their posterior distribution, and $Q_j \sim \mathcal{N}(\mu_{q_j}, \sigma_{q_j}^2)$ denote their covariate-informed prior. Our objective is to minimize total KL divergence:

$$\sum_j^N D_{KL}(P_j \mid Q_j) \tag{26}$$

If $\sigma_q^2$ is constant across patients, then total KL divergence is minimized by minimizing the total squared error $\sum_j^N (\mu_{p_j} - \mu_{q_j})^2$.

We see also that total KL-divergence is minimized at the following value of $\sigma_q^2$:

$$\sigma_q^2 = \frac{1}{N} \sum_j^N (\mu_{p_j} - \mu_{q_j})^2 + \sigma_{p_j}^2) \tag{27}$$

## D  SIMULATION PARAMETERS

Each simulated patient had age, height, weight, and sex sampled independently from the following distributions. These variables were used to parameterize their PK/PD simulations.

$$\text{age} \sim \mathcal{N}(54.9, 17.5) \text{ years} \tag{28}$$
$$\text{height} \sim \mathcal{N}(66.6, 4.3) \text{ in} \tag{29}$$
$$\text{weight} \sim \mathcal{N}(88.1, 20.9) \text{ kg} \tag{30}$$
$$P(\text{sex} = \text{Male}) = 0.47 \tag{31}$$
$$P(\text{sex} = \text{Female}) = 0.53 \tag{32}$$

Intervals between observation times were exponentially distributed, with a mean of 10 observations per 24 hours.

$$\Delta t_{j_i} = t_{j_i} - t_{j_{i-1}} \sim \text{exponential}(10/1440) \tag{33}$$

Intervals between opioid administration times were also exponentially distributed, with a mean of 2 instances of opioid administration per 24 hours. 500 mcg of hydromorphone, a standard bolus, were administered at these intervals in simulation, reducing opioid responsiveness to a scalar in our simulation studies, as only hydromorphone was considered.

Simulated patients with fewer than 10 pain score observations and 2 instances of opioid administration in the first 24 hours were excluded from analysis, and a new patient randomly sampled and simulated.

Covariates for each patient were sampled from the standard multivariate Gaussian distribution.

Let $I_d$ be the $d$-dimensional identity matrix:

$$\boldsymbol{c}_j = [c_{j_1}, ..., c_{j_d}] \sim \mathcal{N}(0, I_d) \tag{34}$$

Given $r_x^2$, we have the following conditional distribution for $x_{j_0} \mid \boldsymbol{c}_j$:

$$x_{j_0} \mid \boldsymbol{c}_j \sim \mathcal{N}\left(\sqrt{\frac{5r_x^2}{d}} \sum_{i=1}^d c_{j_i}, 5(1 - r_x^2)\right) \tag{35}$$

For all values of $r_x^2$, the marginal distribution of $x_{j_0}$ is:

$$x_{j_0} \sim \mathcal{N}(0, 5) \tag{36}$$

Given $r_a^2$, we have the following conditional distribution for $\log \boldsymbol{a}_j \mid \boldsymbol{c}_j$:

$$\log \boldsymbol{a}_j \mid \boldsymbol{c}_j \sim \mathcal{N}\left(-5 + \sqrt{\frac{2.25r_a^2}{d}} \sum_{i=1}^d c_{j_i}, 2.25(1 - r_a^2))\right) \tag{37}$$

For all values of $r_a^2$, the marginal distribution of $\boldsymbol{a}_j$ is:

$$\log \boldsymbol{a}_j \sim \mathcal{N}(-5, 2.25) \tag{38}$$

Table 4: Baseline statistics of study cohort

| Statistic | | |
|---|---|---|
| Total number of patients | 21,652 | (100%) |
| Demographics | | |
|   Age (years) | 56.9 | (16.5) |
|   Sex | | |
|     Male | 10,377 | (47.9%) |
|     Female | 11,275 | (52.1%) |
|   Race | | |
|     White | 17,809 | (82.3%) |
|     Black | 1,103 | (5.1%) |
|     Asian | 582 | (2.7%) |
|     Hispanic | 83 | (0.4%) |
|     Other 2,075 | (9.6%) | |
| Clinical Characteristics | | |
|   ASA Status | | |
|     I | 1,534 | (7.1%) |
|     II | 12,104 | (55.9%) |
|     III | 7,663 | (35.4%) |
|     IV | 347 | (16.%) |
|   BMI (kg/m$^2$) | 29.2 | (7.2) |
|   Ambulatory Surgery | 1,224 | (5.7%) |
|   Inpatient Surgery | 20,428 | (94.3%) |
| Surgical Service | | |
|   Orthopedic Surgery | 6,835 | (31.6%) |
|   General Surgery | 3,774 | (17.4%) |
|   Urology | 2,503 | (11.6%) |
|   Neurosurgery | 1,606 | (7.4%) |
|   Gynecology | 1,350 | (6.2%) |
| Procedural Severity Score | | |
|   Morbidity | 36.3 | (17.6) |
|   Mortality | 55.8 | (33.9) |
| 30-day Mortality | 95 | (0.4%) |
| 30-day Readmission | 2,303 | (10.6%) |
| Surgical duration (hrs) | 1.6 | (1.0, 2.6) |
| PACU length of stay (hrs) | 1.9 | (1.4, 2.5) |
| Inpatient hospital length of stay (days) | 2.4 | (1.4, 5.2) |
| Elixhauser comorbidity index | 2 | (0, 8) |
| Opioid naivety | 17,024 | (78.6%) |

