# OpenReview forum: "Covariate-informed continuous-time gray-box modeling to identify responsiveness of post-surgical pain to opioid therapy"
_ICLR.cc/2025/Conference — Submitted to ICLR 2025_

### Official Review · Reviewer_oqAU · 2024-11-03

**Soundness:** 2
**Presentation:** 2
**Contribution:** 1
**Rating:** 3
**Confidence:** 4

**Summary:**

This paper proposes a "gray box" model (partly straightforward to interpret and partly black box) that quantifies responsiveness of pain to opioid therapy. To demonstrate the effectiveness of the approach, the authors experimented on simulated data as well as a real-world observational study.

**Strengths:**

- The specific application (looking at how different opioid drugs impacts perceived pain) is well-motivated.
- The high-level ideas of the paper are mostly easy to understand/follow (perhaps in part because from a technical standpoint, I find that the proposed method is largely just piecing together fairly standard techniques).
- The proposed approach looks promising.

**Weaknesses:**

- I think this paper would really benefit from having baselines, even if the baselines are "straw man" baselines, just to give the readers a sense of how well much simpler "naive" approaches to solving the problem do. I don't know this specific application well enough to know whether there are any well-known existing baselines or well-known clinical guidelines that would help us better understand how the proposed method compares to what best practices currently are. I understand that the paper shows that the proposed method is able to recover findings consistent with existing literature, but it seems from reading lines 369-375 (page 7) that the method also has some discrepancies with existing literature? More thorough discussion of this discrepancy would be helpful.
- In the related works section, causal inference based approaches are mentioned (such as the Liu et al (2023b) and Bica et al (2020) references). Could these be used as baselines? I think importantly, even if the causal assumptions are not satisfied, it is still worth trying out these models to get a sense of whether they provide anything useful even at the level of quantifying *association* rather than causation.
- I think better justifying the different components of the proposed method would be helpful, especially since I get the impression that many different models could have been developed to solve this particular problem. For example, how much do the results change as we change the black box predictors used? Also, maybe I missed it but I didn't understand which specific black box predictors are used. For the continuous state space part, I'm under the impression that a number of authors have worked on methods in this space that could potentially be applied to your setting as well (for example, some older papers here would be the deep Kalman filter paper by Krishnan et al (2015) or the paper on structured variational autoencoders (Johnson et al 2016); more recently there has been an explosion of papers recently on state space modeling using S4/Mamba architectures, and I'm not sure to what extent those could be applied in your setup).

Minor:
- Some of the math notation is not standard and should be fixed, especially how functions are specified. For example, in the first two paragraphs of Section 3.1, "$y_j(t_{j_i}): \mathbb{R}\rightarrow\\{0,1,\dots,10\\}$" should instead be written as "$y_j: \mathbb{R}\rightarrow\\{0,1,\dots,10\\}$" and "$\boldsymbol{u}_j(t):\mathbb{R}\rightarrow\mathbb{R}^m$" should instead be written as "$\boldsymbol{u}_j:\mathbb{R}\rightarrow\mathbb{R}^m$". Etc. (Note that I can't figure out how to get "\mathbf" to work in OpenReview so I didn't get the bolding to match the text.)

References:
- Krishnan et al. Deep Kalman Filters. NeurIPS Advances in Approximate Bayesian Inference & Black Box Inference (AABI) Workshop 2015.
- Johnson et al. Composing graphical models with neural networks for structured representations and fast inference. NeurIPS 2016.

**Questions:**

Please address the weakness points that I raised (which are fundamentally about *baselines* and a *more thorough literature review* that better justifies the specific modeling choices made). If some modeling choices could actually be swapped out for something else or changed, giving the reader a sense of how much the results change would be helpful.

More generally, especially as I find that this paper is more of an applied paper, I think the paper should more thoroughly interpret the results of the model in the context of the actual application. Being very clear about how the proposed method could be used by clinicians/practitioners and how it compares to what they already are currently doing would be helpful.

---

### Official Review · Reviewer_WaQ2 · 2024-11-03

**Soundness:** 3
**Presentation:** 2
**Contribution:** 2
**Rating:** 5
**Confidence:** 3

**Summary:**

This paper proposed a continuous-time state-space model for quantifying patient responsiveness to opioid therapy by using  pain scores, PK/PD models, and patient covariates. The model uses Bayesian inference and MCMC methods to estimate latent pain states and individual opioid response parameters. A simulation study and real-world data from over 21,000 surgical cases were used to validate the effectiveness of model.

**Strengths:**

1.This paper addresses a interesting and meaningful topic in the AI for healthcare field, focusing on quantifying patient responsiveness to opioid therapy, which is crucial for reducing risks associated with opioid use. Additionally, the authors introduce an novel approach by employing continuous-time state-space modeling, which captures the dynamic nature of pain and drug effects more effectively.

2. The authors provided R code for the model and simulation study, which is helpful for reproducibility.

**Weaknesses:**

[Presentation] The introduction does not clearly outline the basic modeling of the responsiveness of postsurgical pain and the limitations of existing methods, which makes it harder for readers to fully grasp the motivation behind the study. For example, the introduction only discusses the importance and challenges of personalized opioid responsiveness but does not mention any existing work or their limitations. If there are existing or similar studies, please add them to the introduction to provide context and show how your approach differs. The "outcomes" column in the Table 3 are not well-explained, making it difficult for readers to follow


[Method] The proposed method assumes covariate-informed priors, which may strongly impact predictive performance and potentially introduce errors. Additionally, opioid ECS  u_j  values are derived from patient demographics, overlapping with covariates c_j  and potentially introducing correlation issues that could affect model reliability. I recommend that the authors perform specific analyses, such as correlation or multicollinearity tests, to assess the relationship between these variables and evaluate their impact on model outcomes to ensure unbiased predictions.

[Method] This paper employs a complex continuous-time state-space model and stochastic differential equations, which may be hard for clinicians to understand and apply in practice.  Additionally, I recommend that the authors include a more detailed discussion on how the predicted opioid responsiveness can be effectively translated into intervention strategies or treatment plans, providing clearer guidance on the clinical implications and real-world application.

[Experiment] Although the experimental section demonstrates the effectiveness of the model through simulations and real-world data, it lacks evidence or experimental results to support how the proposed method outperforms existing models [1]

[1] Estimating individual treatment effect: generalization bounds and algorithms. ICML2017

**Questions:**

see weakness

---

### Official Review · Reviewer_Lu4G · 2024-11-03

**Soundness:** 3
**Presentation:** 3
**Contribution:** 3
**Rating:** 5
**Confidence:** 4

**Summary:**

The authors propose a general approach to combine black-box ML with gray-box modeling components informed by prior domain knowledge. They apply this to a challenging (non-standard) clinical problem, namely the identification of post-operative patient responsiveness to pain medication. To this end, they combine a continuous-time dynamical state-space model of the patient's latent pain state with a black-box ML model component that adjusts each patient's prior on medication responsiveness based on available covariates. MCMC using a custom proposal function and expectation maximization are used for inference. The approach is validated using a simple simulation study before being applied to a large-scale real-world dataset. The identification results appear to loosely correlate with prior results known from the medical literature.

**Strengths:**

- An interesting, non-standard clinical application problem, a custom solution well-tailored to this particular problem, and a validation on real-world clinical data for this particular problem
- A novel (to me, at least), interesting, uncertainty-aware and quite general way of combining black-box ML with traditional models based on prior domain knowledge (in this case, on pharmacodynamics and -kinetics)
- The paper is generally very well-written and nicely readable; the presented (quite complex) modeling approach is presented well; math is presented thoroughly and precisely
- A thorough approach for MCMC and EM-based inference in the proposed model
- Great related works section, providing a concise yet very helpful overview of various (very different) strands of related literature

**Weaknesses:**

1. I am not yet convinced that the identification was actually successful, yields meaningful results, and is useful in any real way. Two specific points in case:
  - Did the MCMC procedure actually converge? No standard MCMC details or diagnostics are provided. How many chains were used? How many steps? Did they all converge to the same distribution? What are the effective sample sizes (ESS) and $\hat{r}$? What do the trace plots and ACFs look like? Like most inference procedures, MCMC always yields *some* result but it is rarely trustworthy. As a point in case, the responsiveness posteriors in Fig. 3 look like they are strongly dominated by some relatively uninformative prior; they are very much *not* concentrated around a specific parameter value (and characterizing them by the median does not seem to make a lot of sense to me).
  - The identified opioid responsiveness correlates with clinical outcomes (table 3). However, is this correlation actually any better than a much more naive approach such as simply categorizing patients based on the mean reported pain in the first 24h after surgery, ASA status, procedural severity, age, etc.? Do we *gain* anything from using this quite complex approach? In any case, this is only *very* circumstantial evidence that the identified parameters indeed bear any meaningful relationship to real patient properties. (Also, the distributions in Fig. 4 are really not bimodal at all, hence a categorization into high/low groups makes little sense. It would seem much more meaningful to assess *correlations* between the identified parameters and the relevant outcomes instead.)

2. I am not (yet) entirely convinced by the choice of the dynamical model. Eq. (2) suggests that drug administration pushes down the latent pain state, even far into negative territory and even if pain is already suppressed to zero. This seems to neglect e.g. saturation effects and might lead to the latent pain state taking an unrealistically long time to 'recover' back to normal. (I would rather have expected a multi-state model, e.g. with separate latent pain and opioid effect states and pain observations representing the difference of the two.) It also seems unlikely that the state transition noise is actually white (Wiener) - e.g. after stopping opioid administration, the pain state will likely continually increase, no? So I would expect the process noise to be (auto-)correlated.

**Questions:**

My most important questions are already listed above under 'weaknesses'.

In addition, a few minor things:
- What are the actual models $f$ and $g$ used in the case study?
- How is all of this implemented? What are the key packages used for modeling / inference?
- What is the exact motivation for this work? The first paragraph concludes by stating that "Tools to identify patients for whom opioids may be less effective for pain relief and that may have greater risk for dependence are greatly needed." What clinical benefit exactly would such tools enable? In other words, how could the insights derived from such a model be turned into improved medical care?
- The authors write that "Typically, expectation-maximization procedures are used to fit state space models." While possibly true (that EM is 'typically' used), this seems a bit reductive to me? A modern approach might be e.g. to use current autodiff packages and implement maximum likelihood estimation via SGD, see e.g. https://github.com/probml/dynamax ? Särkkä and Svensson, Bayesian Filtering and Smoothing, might be interesting for the authors if they don't know it already (which I assume they do).
- (Black-box) Variational Inference could be listed (and discussed) as a potential alternative to the MCMC approach pursued here
- I found the example presented in Fig.1 to be a bit perplexing, since it actually looks as if opioid effect site concentration does not have a meaningful effect on pain scores at all? Or am I misinterpreting something here? Pain rises and then drops again around 400 min without opioid administration, and then oxycodone is administered twice with no apparent effect at all? What is the reader to take from this figure?
- The acronym 'PACU' is never spelled out / defined. I presume the same holds for several other acronyms.

---

### Official Review · Reviewer_eE31 · 2024-11-06

**Soundness:** 2
**Presentation:** 4
**Contribution:** 2
**Rating:** 3
**Confidence:** 2

**Summary:**

The paper defines a mechanistic Bayesian model that can infer a posterior over patient specific (although patient specific in an incredibly limited sense) opioid responsiveness to opioid treatments from observations of reported pain.

**Strengths:**

1. The model follows Clinical Intuition
The authors the main clinical takeaway that opioid responsiveness is associated with better overall outcomes, and demonstrate this is the case with several pain and risk outcomes. Additionally, the model successfully learns known relative potencies between different opioids (fentanyl vs hydromorphone vs oxycodone) from the clinical literature, which helps validate their approach.

Additionally, mechanistic models are interpretable and practical for a clinician understanding and aiding their decision-making.


2. The model leverages known latent dynamics:
The paper leverages a pharmacology model to estimate opioid concentrations in the patient over time $u(t)$. This domain-specific bias could give this method a huge edge over black box models (especially on this limited dataset size) however the authors do not compare to any black box baselines, or try ablating this from the model and just using raw dosage information.

The model additionally provides an interpretable latent pain score, that ideally is more objective than reported pain which is very noisy.

3. Validation
The authors perform a simulated study demonstrates they can recover the opioid responsiveness for synthetic patients given a 24-hour trajectory. Additionally, they demonstrate the model learns relative potencies known in the medical literature.

**Weaknesses:**

## Key Weaknesses

1. Impact of this application is currently weak
This is mainly because the paper requires a full 24-hour trajectory and provide retrospective insights rather than actionable predictions:

a. The method can only determine the opioid responsiveness **after** seeing the full 24-hour trajectory. I think there should be analysis of the potential impacts and interventions this could enable physicians to make real time decisions based on the inferred patient opioid responsiveness, at different time scales (say 1 hour, 4 hours, 8 hours, etc.)

b. Additionally, patient trajectories are inferred independently. The only cross patient learning is limited to the "covariate-informed prior", a neural net which only take static patient information as input (demographics and a small set of patient clinical characteristics). I imagine there is useful initial trajectory information that is not captured in that limited prior data. More specifically, can we take the first hour or so of the patient's observed dynamics and leverage this for a more informative prior in the model? There must be characteristics of initial responses to treatment that are shared across patients too, and this approach would enable this.

Isn't the result that opioid responsiveness is associated with better overall outcomes confounded by the fact that acutely sicker patients (and chronically sick patients) have more pain and may be flagged as having low responsiveness by your method. A patient may have high pain regardless of the amount of opioids and responsiveness if their physical state is constantly deteriorating. The discussion of these findings and confounders should be significantly deeper.

If a patient is correctly identified as a low responder, what do we do? Alternative remedies are not proposed or analyzed for aiding clinical decision-making.

2. Weak Evaluation -- No baselines or ablations

The paper's proposed model needs a comparison against simpler approaches to justify (a) its added complexity and (b) the additional computational cost. I don't know what an appropriate baseline is, but I would expect there are simple autoregressive baselines such as:
```
For each patient:
1. Take hourly bins of:
   - Average pain score (carry forward and backward impute these pain scores)
   - Total opioid concentration from u(t) and/or the raw opiod dosage in that hour
2. Fit simple linear model:
   pain(t) = β0 + β1*pain(t-1) + β2*drug_concentration(t) + ε
3. Use β2 as estimate of opioid responsiveness $a$
```


You should compare to standard medical assessments of opioid tolerance or physician's labeled estimate to validate your model as well, or demonstrate the specific failures of these standards that your model overcomes.

The authors do not ablate any parts of the model to demonstrate that they are indeed necessary for predicting $a$.


## Presentation issues

Define $\mathcal{L}$ before it is first used. I think you use it as probability, right, why not use $P$?

**Questions:**

1. Why not provide baselines and ablations of your method to defend the design choices and that the added complexity is necessary for improved predictions of $a$? Why wouldn't a simple autoregressive model work pretty well, while maintaining a similar level of interpretability?
2. To improve the impact of this work, please address how would $a$ be used by a clinician, what kind of decision-making can this enable, and demonstrate the efficacy of potential interventions. Can you demonstrate either a case study or an improved causal treatment effect based on some interventions your method allows?
3. Can you remove the restriction of a full 24-hour trajectory and find the limits of your method in terms of how much time it really needs to get an accurate measurement of $a$, and discuss how that could affect clinician decision-making abilities. Specifically, can you get an estimate of $a$ in a short enough time to allow clinicians to make meaningful interventions on choice of opioid.
4. It seems that the result that Opioid responsiveness is associated with better overall outcomes is confounded by the fact that those patients are just sicker. Can latent sickness be included in the model to remove this confounding of severity of the disease, from the patient's responsiveness to opioids.

---

### Meta-Review · Area_Chair_6Li6 · 2024-12-20

**Metareview:**

The paper presents a method to learn opoid responsiveness (measured through pain levels) of post-surgical patients using a state-space model with PKPD domain knowledge information. The results are first validated through a simple simulation study, and the real-world data results loosely correlate to known clinical findings.

Strengths (based on reviewers' input):
 - The application is a novel clinical problem

Weaknesses
 - Identification results are not convincing. In particular, the evaluation details lacked nuance, meaning it is not clear that this model would be impactful
 - Lack of details about the MCM procedure meant that basic information such as the number of steps and number of chains was missing. The posteriors in Fig 3 appear to be dominated by the uninformed prior.
 - Lack of baselines such as categorizing patients based on rough categories or causal inference methods or more similar methodological comparisons like Deep Kalman Filters
 - Unclear motivation for this particular formulation of the problem

Due to the large list of consistent weaknesses spotted by all reviewers, I am recommending reject for this paper. I hope that with additional explanations and baselines, the work will be more mature. I look forward to seeing it featured in another research venue.

**Additional Comments On Reviewer Discussion:**

The authors did not include a response.

---

### Decision · Program_Chairs · 2025-01-22

Reject